# Inhibiting Rho kinase promotes goal-directed decision making and blocks habitual responding for cocaine

Andrew M. Swanson[1,2], Lauren M. DePoy[1,2] & Shannon L. Gourley[1,2]

The prelimbic prefrontal cortex is necessary for associating actions with their consequences, enabling goal-directed decision making. We find that the strength of action–outcome conditioning correlates with dendritic spine density in prelimbic cortex, suggesting that new action–outcome learning involves dendritic spine plasticity. To test this, we inhibited the cytoskeletal regulatory factor Rho kinase. We find that the inhibitor fasudil enhances action–outcome memory, resulting in goal-directed behavior in mice that would otherwise express stimulus-response habits. Fasudil transiently reduces prelimbic cortical dendritic spine densities during a period of presumed memory consolidation, but only when paired with new learning. Fasudil also blocks habitual responding for cocaine, an effect that persists over time, across multiple contexts, and depends on actin polymerization. We suggest that Rho kinase inhibition promotes goal-oriented action selection by augmenting the plasticity of prelimbic cortical dendritic spines during the formation of new action–outcome memories.

[1] Departments of Pediatrics and Psychiatry, Emory University School of Medicine, 954 Gatewood Road NE, Atlanta, GA 30329, USA. [2] Yerkes National Primate Research Center, Graduate Program in Neuroscience, Emory University, 954 Gatewood Road NE, Atlanta, GA 30329, USA. Correspondence and requests for materials should be addressed to S.L.G. (email: shannon.l.gourley@emory.edu)

The ability to select actions based on expected consequences is critical for survival. Such actions are considered "goal-directed," meaning their performance is sensitive to changes in the relationship between the action and its outcome. With repetition, goal-directed behaviors can shift from being outcome-sensitive to being "habitual," which is defined by insensitivity to changes in the action–outcome relationship. Moreover, a range of pathological stimuli, including cocaine, can also bias response strategies to favor habit-based responding[1–9]. Although the anatomical connections that organize goal-directed decision making on the one hand, and habit-based behaviors on the other, are becoming clearer[10], underlying cellular and molecular mechanisms are less well-understood. Further, restoring goal-directed action selection after habits have formed has proven particularly challenging. This is important because although the development of habits can be a normal, adaptive process, habit-based stimulus-elicited reward seeking that occurs at the expense of engaging in goal-directed response strategies may have an etiological role in the development and maintenance of drug addiction and other psychiatric illnesses such as obsessive-compulsive disorder[11].

The prelimbic prefrontal cortex is necessary for goal-directed action selection. For example, prelimbic cortical inactivation impairs the ability of rats to develop and update instrumental response strategies[12–14]. Inactivation after learning has occurred has no effect[15], however, indicating that the prelimbic cortex is necessary for forming, but not necessarily expressing, information regarding the predictive relationship between actions and their outcomes. First, we expand on these findings by showing that chemogenetic inactivation of the prelimbic cortex of mice similarly impairs an animal's ability to select actions based on their consequences.

Prelimbic cortical neurons receive projections from subcortical areas that are likely involved in developing goal-directed response strategies[10, 16]. We thus hypothesized that the structural plasticity of these prelimbic cortical neurons may have an important role in generating goal-directed actions. To test this hypothesis, we inhibited a major substrate of the RhoA GTPase, Rho kinase (ROCK), which stabilizes filamentous (F)-actin[17]. We used the brain-penetrant ROCK inhibitor fasudil, which has been shown to enhance hippocampal- and prefrontal cortical-dependent learning and memory in other contexts[18–21]. We find that fasudil enriches action–outcome conditioning, resulting in goal-directed response selection in mice that would otherwise be expected to express food- or cocaine-seeking habits. We also discovered that individual differences in decision making strategies in mice correlate with dendritic spine densities on prelimbic cortical neurons. Specifically, lower densities are associated with action–outcome-based, goal-directed behavior, with higher densities associated with stimulus-response habits. This correlation raises the possibility that stimulating activity-dependent dendritic spine pruning within the prelimbic cortex might help to restore goal-directed action selection after habits develop. Indeed, we find that ROCK inhibition results in a transient pruning of prelimbic cortical dendritic spines, which appears to be necessary for enhancing action–outcome conditioning. Taken together, our findings suggest that ROCK inhibition may be an effective tool in promoting goal-directed decision making in therapeutic contexts, e.g., for cocaine use and abuse.

## Results

**Prelimbic inactivation impairs action-consequence memory.** To begin these investigations, we utilized an action–outcome contingency degradation procedure (Fig. 1a). In this task, modestly food-restricted mice are trained to generate two food-reinforced operant responses equally, then the likelihood that one

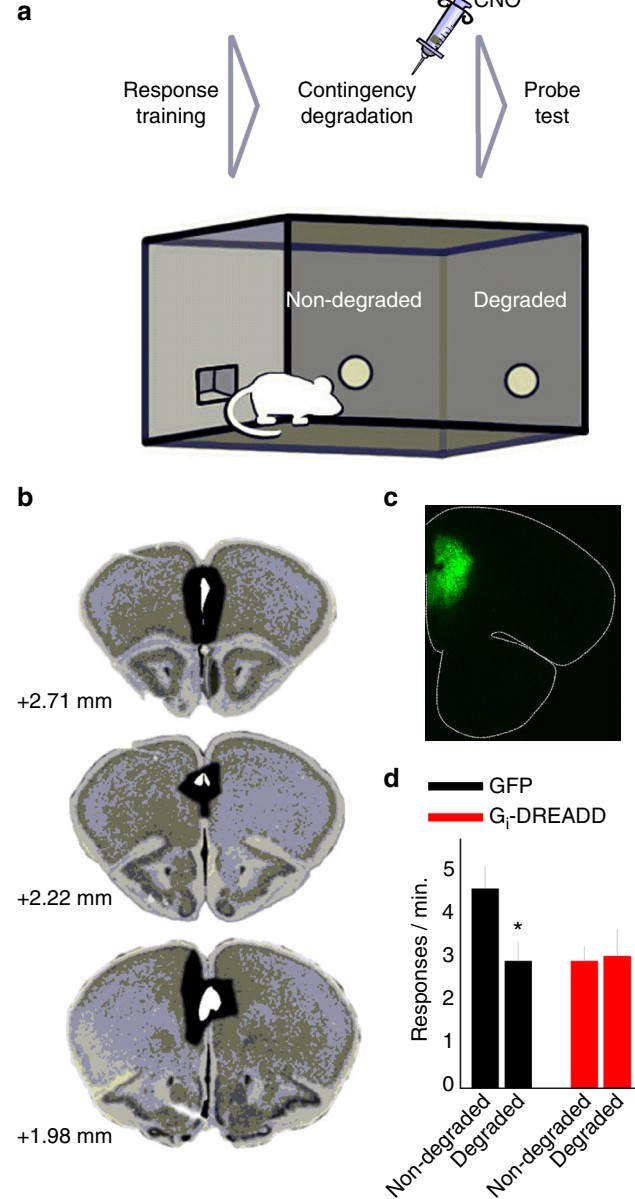

**Fig. 1** Prelimbic prefrontal cortex inactivation impairs the ability of mice to select actions based on their consequences. **a** Experimental timeline: Food-restricted mice are placed in operant-conditioning chambers and trained to respond on two apertures for food pellets. Following training, responding on one aperture continues to be reinforced ("non-degraded"), whereas food pellets associated with the other response are delivered non-contingently, "degrading" the action–outcome contingency. During a probe test, preferential engagement of the behavior that is most likely to be reinforced is considered goal-directed, whereas engaging both responses non-selectively is considered a failure in action–outcome conditioning, a deferral to familiar habit-based behavior. **b** The largest and smallest viral vector spread is represented on images from the Mouse Brain Library[70]. **c** Representative GFP expression is also shown. **d** Following instrumental contingency degradation and CNO injection, control GFP-expressing mice (n = 12) preferentially engaged the response most likely to be reinforced. Gi-DREADDs-expressing mice did not differentiate between the responses, deferring instead to habit-based responding (n = 7). Bars and symbols = means + s.e.m., *p < 0.05. This experiment was conducted twice, with concordant outcomes; a single cohort is represented. We thank A. Allen for generating the illustration used in this figure

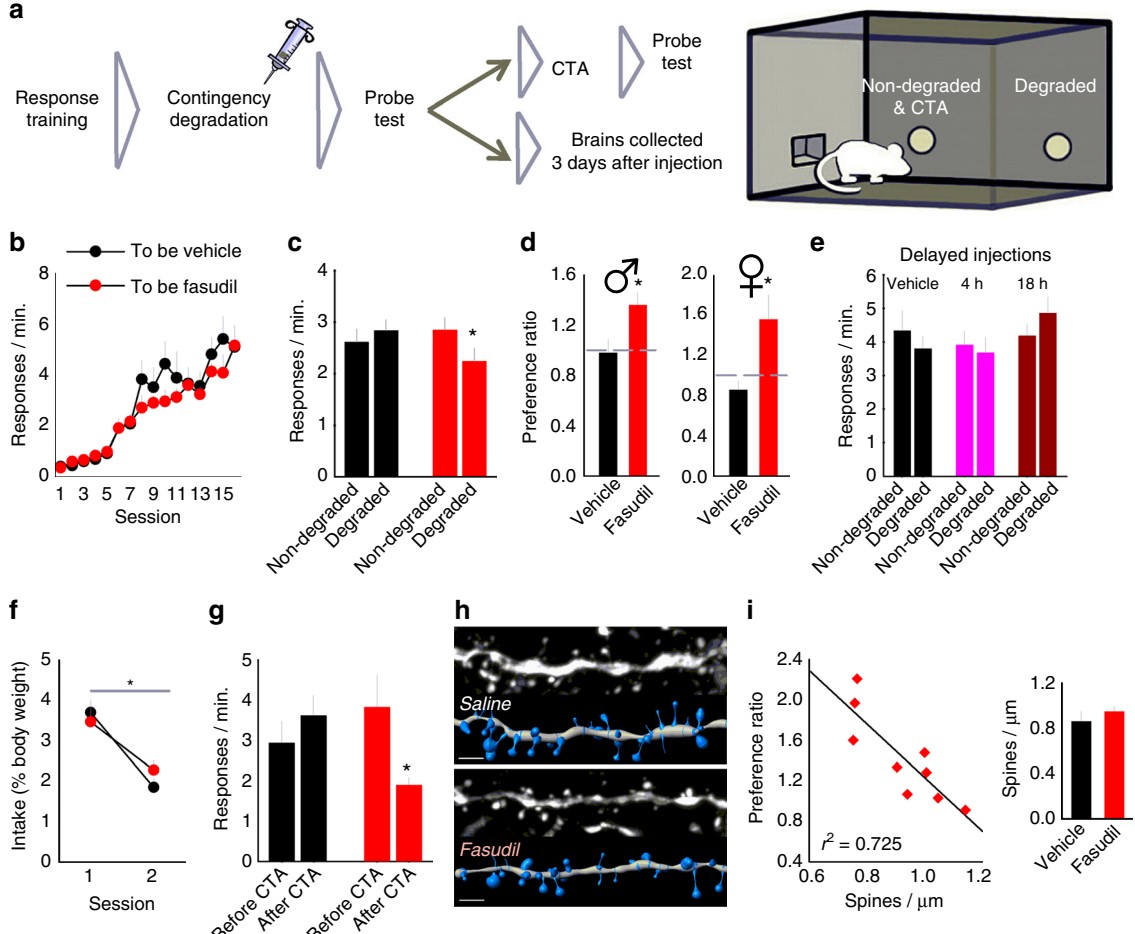

**Fig. 2** ROCK inhibition enhances action–outcome decision making. **a** Experimental timeline: mice were trained using random interval (RI) schedules of reinforcement to induce habit-based behavior. Immediately following instrumental contingency degradation, mice were given a systemic injection of either vehicle ($n = 15$) or the ROCK inhibitor fasudil ($n = 17$). Following a probe test, brains were either collected or mice were tested further for sensitivity to outcome value using lithium chloride (LiCl)-based conditioned taste aversion (CTA). **b** Groups did not differ during instrumental response acquisition. **c** Following instrumental contingency degradation, vehicle-treated mice did not differentiate between the responses that were likely, vs. unlikely, to be reinforced, behaving habitually. By contrast, fasudil-treated mice preferentially engaged the response most likely to be reinforced. **d** The same data are represented as preference ratios (non-degraded/degraded contingencies), with values $> 1$ indicating goal-directed responding. Vehicle-treated mice responded at chance levels (habit), whereas fasudil-treated mice utilized a goal-directed response strategy. Right: although we focus our report on males, we found that female mice were also sensitive to fasudil in the same task ($n = 7$, 8 per group). **e** When fasudil injections were delayed by 4 or 18 h, no groups responded preferentially, instead relying on habitual response strategies ($n = 8$ per group). **f** We next induced CTA ($n = 11$, 12 per group), reducing the amount of food consumed over two pairings. **g** Despite CTA, previously vehicle-treated mice failed to reduce responding, consistent with habitual behavior following instrumental contingency degradation (in **c**). Meanwhile previously fasudil-treated mice reduced responding, sensitive to the now-reduced value of the reinforcer. **h** Representative prelimbic cortical dendritic segments with corresponding digital reconstructions below. **i** Dendritic spine densities correlated with response selection strategies, although spine densities did not differ between groups (inset). Scale bar $= 2\,\mu m$. Bars and symbols $=$ means $+$ s.e.m. except in **i** where diamonds represent individual mice, $*p < 0.05$. For each experiment, $> 2$ cohorts of mice were tested. We thank A. Allen for generating the illustration used in this figure

response will be reinforced is reduced; instead, food pellets associated with that response are provided non-contingently. Meanwhile, the other response remains reinforced. In a probe test conducted in extinction, both response operandi are available, and preferential engagement of the behavior that is likely to be reinforced provides evidence of knowledge of the action–outcome relationship, whereas a failure to modify response strategies reflects a deferral to familiar, habit-based behaviors[22].

To confirm that the prelimbic cortex in mice is involved in action–outcome-based decision making, we delivered viral vectors expressing either CaMKII-driven GFP or G$_i$-coupled Designer Receptors Exclusively Activated by Designer Drugs (DREADDs[23]) to the prelimbic cortex (Fig. 1b, c). Following recovery, all mice acquired the food-reinforced operant responses,

with no differences between groups (Supplementary Fig. 1). Given that the prelimbic cortex is necessary for forming, but not expressing, new memory regarding the predictive relationship between actions and their consequences[15], G$_i$-DREADDs were next pharmacologically activated in conjunction with action–outcome contingency degradation. Then, mice were subsequently tested for response preferences during a probe test, drug-free. Control GFP-expressing mice preferentially performed the response most likely to be reinforced in a goal-directed fashion, whereas G$_i$-DREADDs-expressing mice responded non-selectively, deferring to a habit-based strategy (ANOVA: interaction $F_{(1,17)} = 4.370$, $p = 0.05$) (Fig. 1d). Thus, chemogenetic silencing of the prelimbic prefrontal cortex impaired action–outcome learning and memory.

**Dendritic spine densities predict decision making strategies**. Multiple types of learning and memory are thought to require activity-dependent dendritic spine proliferation on the one hand, or dendritic spine elimination, which can also be activity-dependent, on the other[24]. To dissect whether dendritic spine dynamics influence action–outcome learning and memory, we next manipulated the cytoskeletal regulatory factor ROCK, a major substrate of the RhoA GTPase that significantly impacts activity-dependent structural remodeling[25, 26].

In this experiment, mice were given extensive response training using random interval (RI) schedules of reinforcement to induce habit-based responding, which is by definition insensitive to the predictive relationship between an action and an outcome[27] (Fig. 2a). Groups (to be vehicle vs. to be fasudil) were designated by matching response rates during training (ANOVA: no interaction $F_{(15,450)} = 1.217$, $p = 0.255$, main effect of session $F_{(15,450)} = 108.956$, $p < 0.001$, no effect of group $F < 1$) (Fig. 2b).

We hypothesized that neuronal structural plasticity could be associated with the formation of new memory, so we administered fasudil immediately following instrumental contingency degradation, during the presumptive period of new memory consolidation. During the probe test the following day, the vehicle group showed no response preferences, i.e., habit-based responding, as expected. In contrast, the fasudil group preferentially generated the response that was likely to be reinforced, a goal-directed strategy (ANOVA: interaction $F_{(1,30)} = 4.560$, $p = 0.04$) (Fig. 2c). Comparing response preference ratios revealed that the fasudil group responded well above chance (habit) levels in its preference for the behavior that was most closely associated with reinforcement, whereas vehicle-treated mice generated non-selective, habitual response strategies, at a ratio of 1 (Mann–Whitney $U = 53$, $p = 0.005$) (Fig. 2d). Although we focus in this report on male mice, we also found that fasudil had the same effects in gonadally intact female mice (Mann–Whitney $U = 3$, $p = 0.008$) (Fig. 2d, right).

In this experiment, fasudil was administered immediately following action–outcome contingency degradation, with the hypothesis that ROCK inhibition could enhance the consolidation of action–outcome learning and memory. This hypothesis predicts that fasudil treatment that is unpaired from a learning opportunity should have no behavioral effects. To test this, we trained separate mice to respond for food reinforcement (Supplementary Fig. 2). Next, vehicle or fasudil was delivered, but injections were delayed 4 or 18 h after action–outcome contingency degradation. In a subsequent probe test, no mice showed a preference for the response that was likely to be reinforced (ANOVA: no interaction $F < 1$, no effect of response $F_{(1,42)} = 1.538$, $p = 0.222$, no effect of group $F < 1$) (Fig. 2e). Thus, ROCK inhibition appears to enhance the consolidation of action–outcome conditioning during a < 4-h time window, facilitating subsequent goal-directed action.

To rule out the possibility that the effects of fasudil could be attributable to enhancing extinction, these mice were next given 3 days of extinction training, with vehicle or fasudil administered concurrent with each session. All mice extinguished responding, and fasudil had no within- or between-sessions effects (Supplementary Fig. 2), suggesting that ROCK inhibition enhanced new action–outcome learning, rather than response extinction per se.

Insensitivity to action–outcome contingencies—as in extensively trained mice here—is often accompanied by insensitivity to outcome value[27]. To assess whether ROCK inhibition could also restore value-based responding (and thus, block habits), we used a devaluation procedure. The male mice in these experiments were trained for two additional sessions to reinstate responding and ensure comparable responding between groups. Next, mice were placed individually in an empty cage and allowed free access

to the reinforcer pellets used in instrumental conditioning experiments for 1 h. Immediately following, mice were injected with lithium chloride (LiCl), inducing transient malaise and conditioned taste aversion (CTA), as evidenced by reduced food consumption across two sessions (ANOVA: no interaction $F < 1$, main effect of session $F_{(1,21)} = 16.364$, $p < 0.001$, no effect of group $F_{(1,21)} = 2.080$, $p = 0.164$) (Fig. 2f). When returned to the conditioning chambers drug-free, mice previously treated with vehicle did not reduce responding relative to the last day of training, despite reinforcer devaluation; in other words, control mice responded habitually, as expected. By contrast, a history of fasudil treatment reduced responding, evidence that fasudil-treated mice used the value of the reinforcer to guide their behavior (ANOVA: interaction $F_{(1,21)} = 4.794$, $p = 0.04$) (Fig. 2g). Altogether, these findings further indicate that the ROCK inhibitor fasudil enhances action–outcome conditioning, blocking habits in favor of goal-directed response strategies.

One cohort of mice tested in the instrumental contingency degradation experiment (in Fig. 2b–d) was killed 72 h after injection. *Thy1-YFP* expression in these mice enabled us to image and quantify prelimbic cortical dendritic spines (Fig. 2h). Somewhat surprisingly, densities did not differ between groups (Fig. 2i, inset), nor did dendritic spine length or head diameter (compared by Kolmogorov–Smirnov tests throughout: Supplementary Table 1). However, densities correlated with decision making strategies, such that lower densities were associated with selecting actions that were likely to be reinforced in a goal-directed fashion (linear regression: $r^2 = 0.725$, $p < 0.05$) (Fig. 2i).

**Enrichment of action–outcome memory**. We next determined whether the behavioral effects of fasudil were attributable to efficacy in the prelimbic prefrontal cortex using intracranial fasudil delivery (Fig. 3a, b). Naive mice were trained to nose-poke for food reinforcers, and groups (to be vehicle vs. to be fasudil) were designated by matching response rates (ANOVA: no interaction $F < 1$, main effect of session $F_{(6,144)} = 39.253$, $p < 0.001$, no effect of group $F_{(2,24)} = 1.585$, $p = 0.226$) (Fig. 3c). Here, the training period was shorter because the stress of intracranial surgery would be expected to bias responding towards habits[28]. Immediately following instrumental contingency degradation training, vehicle or fasudil was intracranially infused. Subsequently, vehicle-infused mice generated habit-based response patterns (no preference for the response most likely to be reinforced), as expected. Meanwhile, prelimbic cortical-targeted fasudil infusions blocked habit-based responding, inducing a preference for the response most likely to be reinforced. By contrast, infusions that had unintentionally terminated in the adjacent anterior cingulate cortex had no effects (ANOVA: with control groups combined, interaction $F_{(2,24)} = 1.659$, $p = 0.007$) (Fig. 3d). These findings suggest that inhibiting ROCK within the prelimbic cortex enhances action–outcome conditioning.

ROCK is endogenously suppressed by Abl2/Arg kinase, such that Arg ablation disinhibits RhoA GTPase-ROCK interactions[29] and blocks dendritic spine plasticity in response to stimuli such as cocaine[30]. We obtained *arg* knockout mice (*arg–/–*) and an Abl-family (Abl and Arg) kinase inhibitor, STI-571. Using a training procedure that biases responding in typical mice towards action–outcome-sensitive goal-directed strategies, we found that *arg* deficiency and prelimbic cortex-targeted STI-571 infusions weakened action–outcome learning, inducing a deferral to familiar, habit-based behaviors (ANOVA: $F_{(2,22)} = 6.170$, $p = 0.007$) (Fig. 3e, right; see also Supplementary Fig. 3). Thus, silencing the endogenous ROCK inhibitor Arg produced the opposite behavioral effect relative to the ROCK inhibitor fasudil,

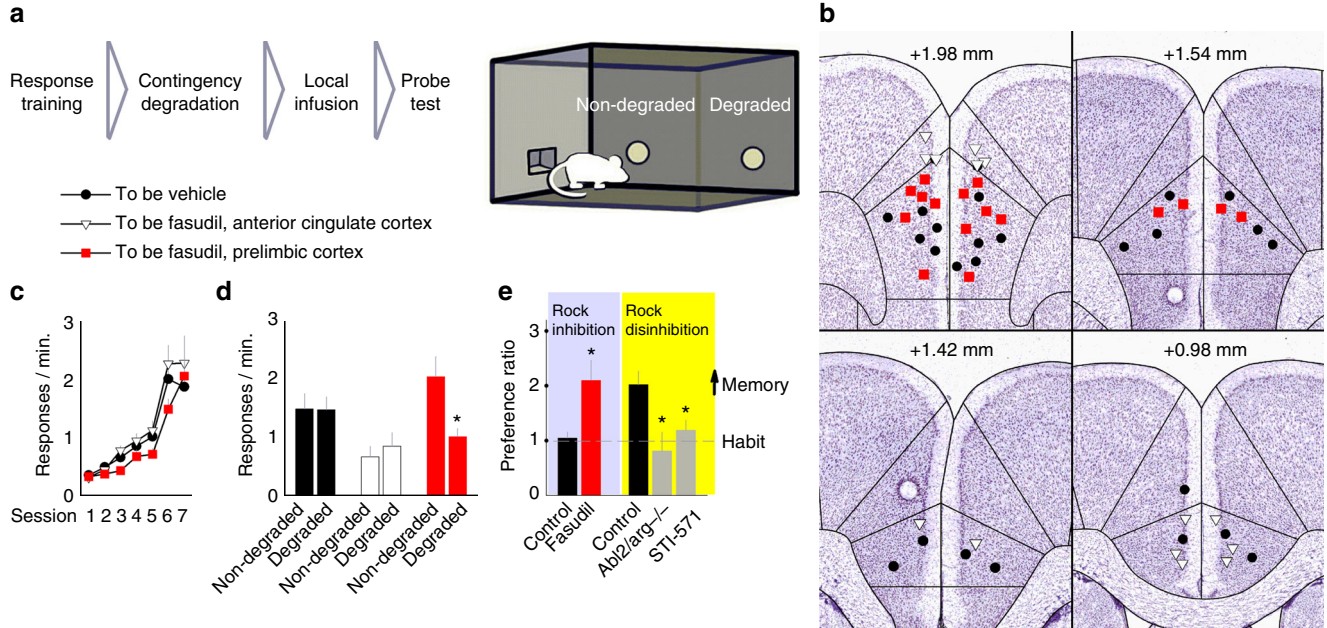

**Fig. 3** Bidirectional coordination of action–outcome memory. **a** Experimental timeline: mice were trained to respond for food reinforcement. Immediately after instrumental contingency degradation, vehicle or fasudil was infused into either the prelimbic or anterior cingulate cortex. **b** Infusion sites, with coordinates relative to Bregma, are indicated on images adapted from the Allen Brain Atlas. **c** All mice acquired the responses, with no differences between groups. **d** Subsequently, vehicle-infused mice failed to differentiate between the responses that were more, vs. less, likely to be reinforced (non-degraded vs. degraded), deferring to habit-based behavior. By contrast, prelimbic cortical infusions of fasudil produced a preference for the response most likely to be reinforced, whereas fasudil in the anterior cingulate cortex had no effects ($n = 12$ combined control, 8 prelimbic, 7 anterior cingulate). **e** The same data are represented as preference ratios (non-degraded/degraded), with values >1 indicating goal-directed responding. Vehicle-infused mice responded at chance levels (habit), whereas prelimbic fasudil-infused mice utilized a goal-directed response strategy. Right: by contrast, in mice trained to be sensitive to action–outcome relationships, knockout of the endogenous ROCK inhibitor *Abl2/arg* or local infusion of an Arg inhibitor STI-571 ablated response preference, indicating impaired action–outcome decision making ($n = 9$ combined control, 6 arg−/−, 10 STI-571). Raw data are reported in Supplementary Fig. 3. Bars and symbols = means + s.e.m., *$p < 0.05$. For each experiment, >2 cohorts of mice were tested. We thank A. Allen for generating the illustration used in this figure

which stimulated goal-directed response strategies (same data as in Fig. 3d represented as preference ratios, compared by Student's *t*-test: $t(18) = 3.311$, $p = 0.004$) (Fig. 3e, left).

**Transient, conditioning-dependent dendritic spine remodeling.** Our findings suggest that mice can use action–outcome relationships to guide decision making strategies, and this process can be facilitated by the ROCK inhibitor fasudil. This could imply that activity-dependent dendritic spine plasticity is associated with action–outcome learning. To test this hypothesis, we again trained mice to respond for food reinforcement, followed by instrumental contingency degradation training and systemic fasudil treatment (Fig. 4a). Groups were designated by matching mice based on response rates during training (ANOVA: no interaction F < 1, main effect of session $F_{(15,225)} = 111.846$, $p < 0.001$, no effect of group F < 1) (Fig. 4b). Instead of testing response preference in a probe test the next day, as in our experiments above, brains were collected 1 h after injection. Dendritic spine imaging and enumeration revealed that fasudil caused a 9% reduction in spine densities on excitatory prelimbic cortical neurons (Student's *t*-test: $t(15) = -2.608$, $p = 0.020$) (Fig. 4c, d; morphology values in Supplementary Table 1), but had no effects on densities in the anterior cingulate cortex (Supplementary Fig. 4a; morphology values in Supplementary Table 1). Further, the densities of prelimbic cortical mushroom-shaped dendritic spines—those likely to contain synapses—were reduced by 16% in the fasudil group (Student's *t*-test: $t(15) = -2.357$, $p = 0.032$) (Fig. 4e). Other spine subtypes (thin and stubby) were not affected (Supplementary Fig. 4b).

We next asked: Is fasudil simply "damaging" neurons? One marker of structural damage is dendritic blebbing, causing the dendrite to enlarge. 3-D dendrite reconstruction revealed that fasudil did not alter dendrite diameter (Student's *t*-test: $t(15) = -0.979$, $p = 0.343$) (Fig. 4f), suggesting that neurons were not damaged.

We next trained another group of mice to nose-poke for food reinforcers (ANOVA: no interaction F < 1, main effect of session $F_{(15,195)} = 119.827$, $p < 0.001$, no effect of group F < 1) (Fig. 4g, h). In this case, fasudil was delayed, administered 1 day after contingency degradation training, with brains again collected 1 h after injection. Here, we found no differences between groups in prelimbic cortical dendritic spine densities (Student's *t*-test: $t(13) = -0.834$, $p = 0.419$) (Fig. 4i, j; Supplementary Fig. 4c), spine length, or head diameter (Supplementary Table 1). Altogether, these findings suggest that fasudil prunes mushroom-shaped prelimbic cortical dendritic spines in an experience-dependent manner.

Notably, dendritic spine densities were higher in these analyses than those in Fig. 2. This may be due to the recentness of injection prior to killing, given that injection stress can trigger dendritic spine proliferation on apical dendrites in the prelimbic cortex[31]. In addition, we preferentially imaged dendritic segments within our imaging window that were distal to the soma, since these dendrites are considered more labile, and thus, potentially more likely to change following drug treatment.

**ROCK inhibition blocks habitual responding for cocaine.** We report here that inhibiting ROCK using fasudil enhances action–outcome learning and memory, blocking habit-based

responding. These experiments were conducted using food reinforcement. To build on this finding, we next assessed whether fasudil can block drug (cocaine) habits as well. Mice were trained to respond on a single operandum for a liquid solution containing cocaine and sucrose that was consumed orally (adapted from ref. [1]) (Fig. 5a). Groups were assigned by matching mice based on response rates during training (ANOVA: no interaction $F_{(16,496)}$ = 1.175, $p = 0.284$, main effect of session $F_{(16,496)}$ = 38.453,

$p < 0.001$, no effect of group $F < 1$) (Fig. 5b). Next, the cocaine solution was paired in a separate context with LiCl, inducing CTA (ANOVA: no interaction $F < 1$, main effect of session $F_{(2,62)}$ = 23.005, $p < 0.001$, no effect of group $F < 1$) (Fig. 5c). Next, mice were briefly returned to the operant-conditioning chambers, providing them with an opportunity to update the association between the cocaine-reinforced response and the now-devalued cocaine[32, 33]. Immediately after this session, mice received an i.p. injection of vehicle or fasudil. During a probe test the next day, the vehicle-treated group continued to respond, indicating habitual behavior. By contrast, fasudil reduced responding, indicating goal-directed, value-based action selection (ANOVA: interaction $F_{(1,31)}$ = 4.322, $p = 0.046$) (Fig. 5d). A post-probe consumption test revealed no differences in cocaine intake between groups (Mann–Whitney $U = 121$, $p = 0.624$) (Fig. 5c) and re-confirmed that both groups acquired the CTA.

Given that systemic fasudil treatment had persistent effects in inhibiting habitual responding for food (Fig. 2f, g), we next assessed whether fasudil could persistently mitigate habitual responding for cocaine as well. We also aimed to confirm that the effects of fasudil in the present experiment could be attributed to sensitivity to the reduced value of cocaine, as opposed to the sucrose included in the cocaine–sucrose solution. To accomplish these goals, a cohort of the mice tested in Fig. 5b–d was surgically implanted with indwelling jugular catheters for intravenous cocaine self-administration. After recovery, mice were trained in a different room, in different operant-conditioning chambers, to respond for intravenous cocaine delivery. Mice acquired the cocaine-reinforced response (ANOVA: main effect of session $F_{(6,96)}$ = 7, $p < 0.001$), however mice with a history of fasudil treatment generated lower cocaine-reinforced response rates (ANOVA: main effect of group $F_{(1,16)}$ = 6.100, $p = 0.025$) (Fig. 5e). Responding on the inactive nose-pokes (i.e., responses that did not result in reinforcement) was also reduced by fasudil, but unlike with cocaine-reinforced response, only during the first session (ANOVA: interaction $F_{(6,96)}$ = 4, $p < 0.001$). Further, the fasudil group required more than twice as many test sessions to ingest 20 mg/kg cocaine in a single session (Mann–Whitney $U = 16.501$, $p = 0.041$) (Fig. 5e, inset). This outcome indicates that ROCK inhibition can enhance sensitivity to cocaine devaluation, reducing cocaine self-administration in an i.v.-delivered setting.

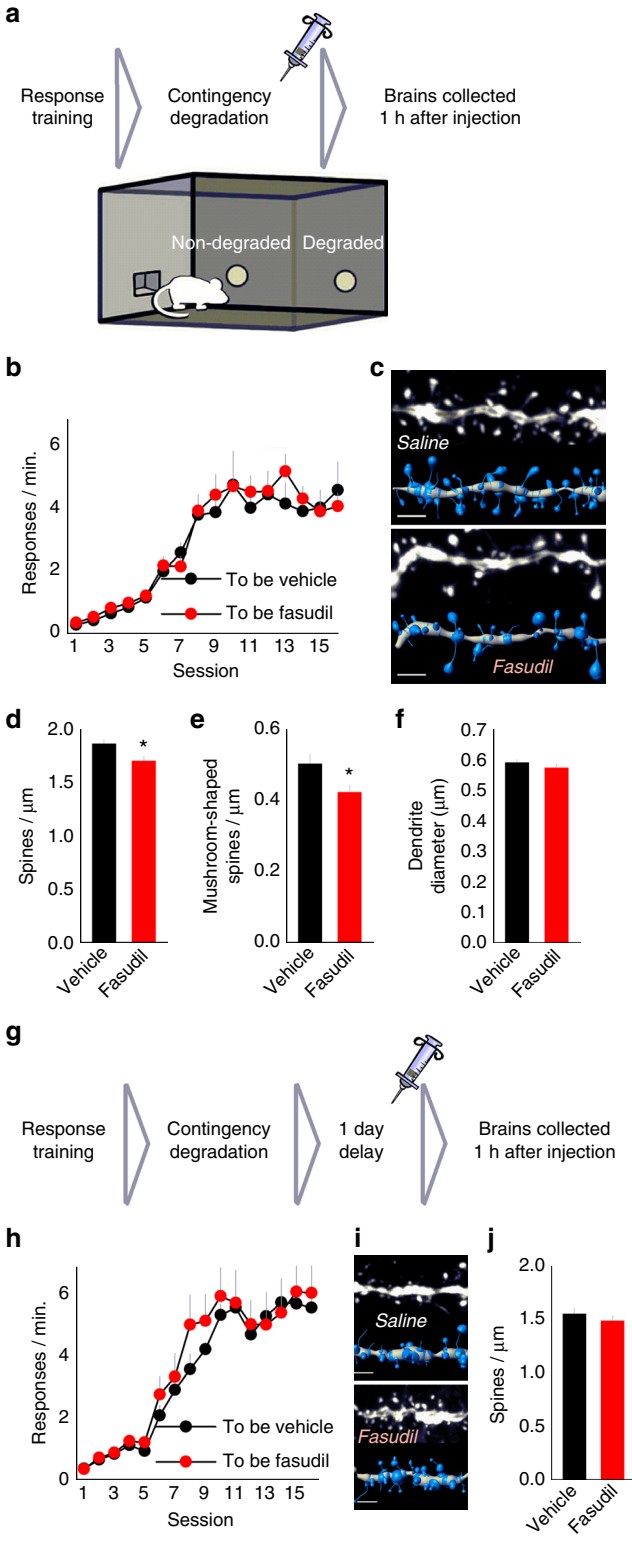

**Fig. 4** Selective remodeling of prelimbic cortical dendritic spines. **a** Experimental timeline: mice were given extensive response training to promote habit-based responding. A systemic injection of vehicle or fasudil was administered immediately following instrumental contingency degradation, with brains collected 1 h after the injection ($n = 8$, 9 per group). **b** Mice acquired the instrumental responses, with no group differences. **c** Prelimbic cortical dendritic segments with corresponding digital reconstructions below. **d** Fasudil reduced prelimbic cortical dendritic spine density. **e** Dendritic spine classification revealed that specifically, mushroom-shaped spines were pruned, with fasudil-exposed segments expressing as few as 0.3 per μm (represented segment above). Densities of all spine types are represented in Supplementary Fig. 4. **f** Dendrite diameter did not differ between groups. **g** Experimental timeline: Another group of mice was trained and tested identically except that injections were administered 1 day after instrumental contingency degradation ($n = 7$, 8 per group). **h** All mice acquired the responses without differences between groups. **i** Representative prelimbic cortical dendritic segments with corresponding digital reconstructions below. **j** A delayed injection of fasudil did not alter prelimbic cortical dendritic spine density, suggesting that fasudil-induced dendritic spine elimination is activity (learning)-dependent. Scale bar = 2 μm. Bars and symbols = means + s.e.m., *$p < 0.05$. We thank A. Allen for generating the illustration used in this figure

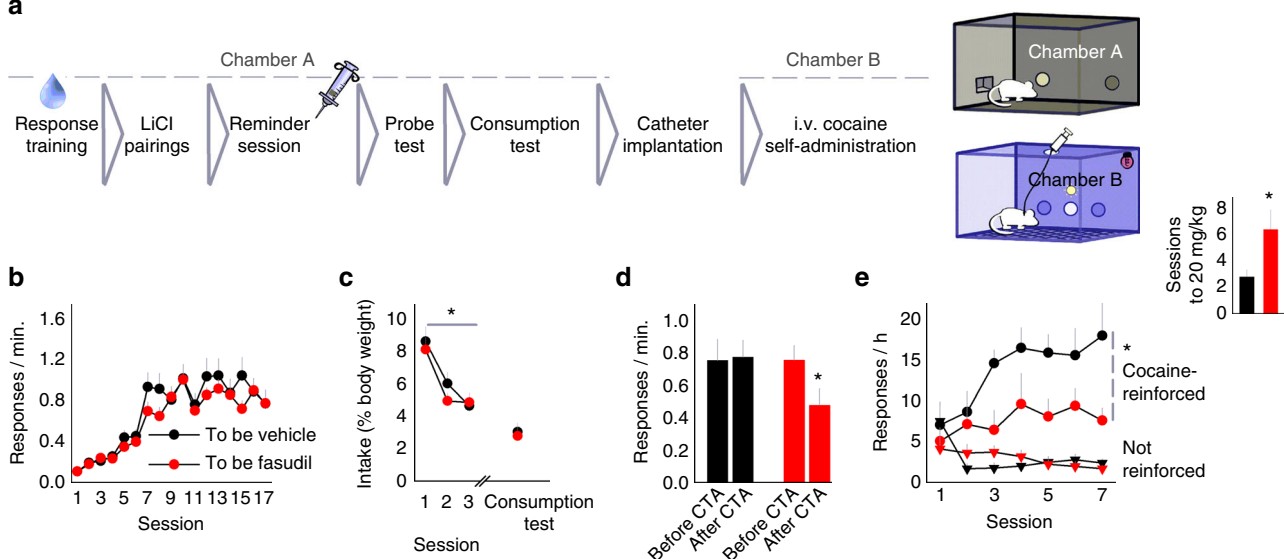

**Fig. 5** ROCK inhibition blocks habitual responding for cocaine. **a** Experimental timeline: Mice were trained to respond for an orally ingested cocaine–sucrose solution. Then, mice were subject to LiCl-induced CTA. Mice were then placed in the conditioning chambers for a brief "reminder session," which served as an opportunity for mice to update the association between the now-devalued outcome and responding. Mice were administered either vehicle or fasudil, followed by a probe test and finally, a post-probe consumption test. These mice were then implanted with indwelling jugular catheters for i.v. cocaine self-administration in contextually distinct conditioning chambers. **b** All mice responded for the cocaine–sucrose solution, without differences between groups ($n = 15, 18$ per group). **c** During CTA, consumption diminished. **d** During a probe test, however, vehicle-treated mice did not reduce responding, despite CTA, indicating insensitivity to the now-reduced value of the cocaine reinforcer. In contrast, fasudil-treated mice reduced responding, indicating sensitivity to outcome value. **e** Mice were trained in contextually distinct chambers to self-administer i.v.-delivered cocaine. Although all mice ultimately acquired the response, mice with a prior history of fasudil treatment responded less throughout and required more sessions to ingest 20 mg/kg (inset). This outcome indicates that fasudil enhanced sensitivity to devaluation of the cocaine reinforcer, and not simply the sucrose that was part of the orally ingested solution. Bars and symbols = means + s.e.m., *$p < 0.05$. Three cohorts of mice were tested. We thank A. Allen for generating the illustration used in this figure

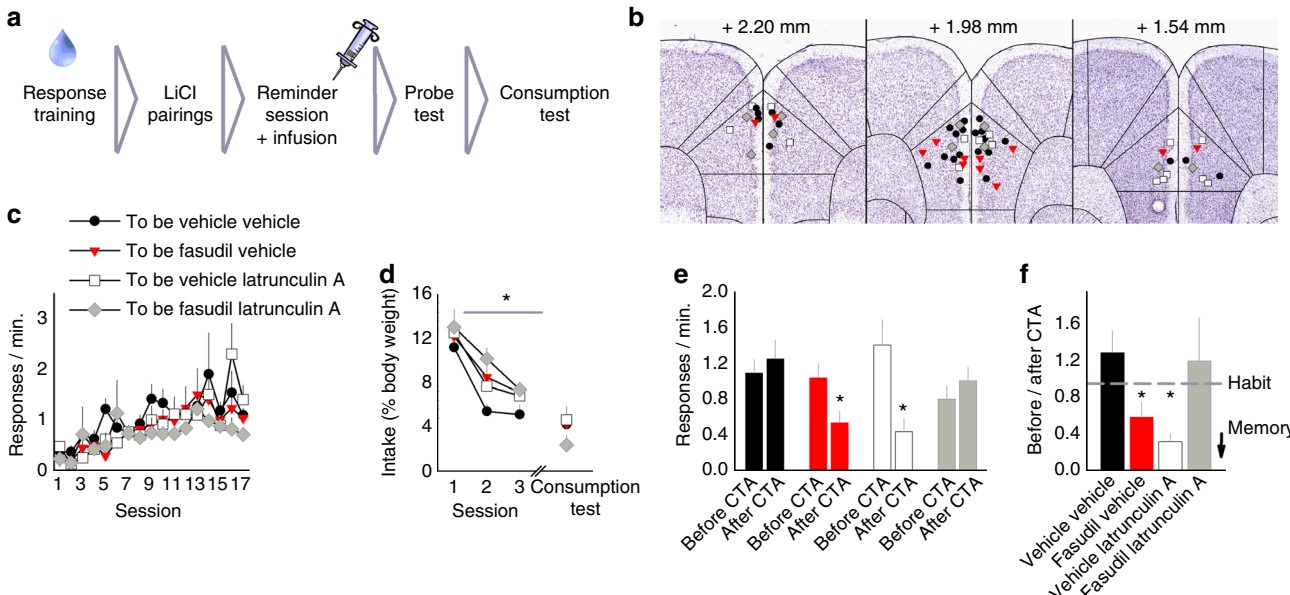

**Fig. 6** Blocking F-actin polymerization in the prelimbic cortex prevents ROCK inhibition from promoting goal-directed action selection. **a** Experimental timeline: As in Fig. 5, mice were extensively trained to respond for an orally ingested cocaine–sucrose solution, followed by LiCl-mediated CTA and a brief "reminder session" paired with vehicle or fasudil. Thirty min after this injection, vehicle or latrunculin A was infused into the prelimbic cortex. Probe and post-probe consumption tests followed. **b** Prelimbic cortical infusion sites are indicated on images adapted from the Allen Brain Atlas. **c** All mice acquired the cocaine-reinforced response, with no group differences. **d** CTA reduced cocaine–sucrose consumption. **e** During the probe test, however, vehicle–vehicle-treated mice did not reduce responding relative to the last day of training ("before CTA"), indicating habitual response strategies. As expected, fasudil–vehicle-treated mice reduced responding, showing sensitivity to the now-reduced value of the reinforcer. This effect was blocked by prelimbic cortical-selective latrunculin A infusion. **f** The same data are represented as the ratio of response rates before vs. after CTA. Scores of 1 represent no change from baseline, whereas scores <1 represent response inhibition (*compared to vehicle–vehicle group) ($n = 13$ vehicle–vehicle, 6 fasudil–vehicle, 8 vehicle–latrunculin A, 6 fasudil–latrunculin A). Bars and symbols = means + s.e.m., *$p < 0.05$. Four cohorts of mice were tested

To address the potential concern that, rather than promoting sensitivity to action–outcome relationships, ROCK inhibition could be having a generalized inhibitory effect on the acquisition of new instrumental behaviors, a cohort of the mice generated in Fig. 5b–d was not subjected to catheter implantation, but instead subsequently trained in different chambers to respond on a different nose-poke aperture for sucrose pellets, a novel reinforcer that had not been devalued (Supplementary Fig. 5a). Groups did not differ in the acquisition of this new sucrose-reinforced response (Supplementary Fig. 5b). Thus, the response-inhibitory effects of fasudil were selectively associated with a devalued reinforcer.

We also assessed cocaine self-administration in mice with a history of food-reinforced responding (mice from Fig. 2b–d). This procedure allowed us to further confirm that fasudil interfered with cocaine self-administration by enhancing behavioral sensitivity to cocaine devaluation, rather than by non-specifically reducing cocaine self-administration in general. Here, a history of fasudil resulted in less responding for cocaine on day 1, likely because nose-poking had recently generated no reinforcement, but this difference was quickly lost (Supplementary Fig. 5c, d). Thus, durable cocaine avoidance results from pairing fasudil with the devaluation of cocaine.

**Effects of inhibiting F-actin polymerization.** ROCK inhibition can increase actin turnover, which is presumably the mechanism by which fasudil eliminates prelimbic cortical dendritic spines and promotes goal-directed decision making. To test this perspective, we sought to disrupt actin turnover with latrunculin A, which blocks F-actin polymerization. In the presence of latrunculin A, fasudil should be ineffective. We again trained mice to respond for an orally available cocaine–sucrose solution, followed by LiCl-induced CTA, and then a "reminder session" paired with a systemic injection of vehicle or fasudil. Vehicle or latrunculin A was then infused intracranially into the prelimbic prefrontal cortex 30 min later (Fig. 6a, b). Subsequently, mice were given a probe test to assess decision making strategies and finally, a post-probe consumption test to re-confirm that all mice acquired the CTA.

Groups were matched based on response rates during training (ANOVA: no interaction F < 1, main effect of session $F_{(16,448)} = 5.813$, $p = 0.001$, no effect of injection by infusion F < 1) (Fig. 6c). Over the course of three pairings of LiCl with the cocaine–sucrose solution, all mice reduced consumption (ANOVA: no interaction F < 1, main effect of session $F_{(2,56)} = 94.125$, $p < 0.001$, no effect of injection by infusion $F_{(1,28)} = 1.301$, $p = 0.264$) (Fig. 6d). A post-probe consumption test re-confirmed that all mice acquired the CTA (ANOVA: no interaction F < 1, no effect of injection F < 1, no effect of infusion F < 1) (Fig. 6d). Nevertheless, when response rates during the probe test were compared against the last day of training, vehicle–vehicle mice were insensitive to the change in outcome value, as expected, whereas fasudil–vehicle-treated mice were sensitive to the change in outcome value and reduced responding in the probe test, replicating the results of Fig. 5d. As predicted, local infusion of latrunculin A into the prelimbic prefrontal cortex blocked the effects of fasudil, resulting in response rates that did not change relative to the last day of training (ANOVA: three-factor interaction $F_{(1,29)} = 6.086$, $p = 0.020$) (Fig. 6e). Surprisingly, mice that received a systemic injection of saline and locally infused latrunculin A were also sensitive to the change in outcome value and reduced responding.

The same pattern can be observed when response rates post-CTA are normalized to those prior to CTA (Fig. 6f). In vehicle–vehicle and fasudil–latrunculin A groups, values were ~1, indicating no response inhibition. By contrast, fasudil–vehicle

and vehicle–latrunculin A groups generated scores < 1, indicating sensitivity to reinforcer devaluation and the suppression of habits (ANOVA: interaction $F_{(1,28)} = 7.694$, $p = 0.01$) (Fig. 6f). These findings support the idea that ROCK inhibition promotes goal-directed action selection in an actin turnover-dependent manner.

## Discussion

Here we trained mice to generate food- or cocaine-reinforced operant responses, then decreased the likelihood that responding would be reinforced or decreased the value of the reinforcer. Response inhibition is thought to reflect new knowledge regarding action–outcome relationships, whereas a failure to modify response strategies reflects a deferral to familiar, habit-based behaviors[27]. We found that inactivation of the prelimbic cortex in mice reduced behavioral sensitivity to action–outcome contingencies, as with prelimbic cortical lesions in rats[12, 13]. In intact mice, lower dendritic spine densities on prelimbic cortical neurons were associated with successful action–outcome learning and memory, reminiscent of evidence that other forms of learning and memory involve dendritic spine pruning[34, 35]. Accordingly, systemic and local prelimbic cortical administration of the ROCK inhibitor fasudil enhanced action–outcome conditioning, blocking habitual responding for food and cocaine in an actin turnover-dependent manner. Fasudil also transiently reduced prelimbic cortical dendritic spine densities during the consolidation of new action–outcome associations. Thus, inhibiting ROCK appears to interfere with habit-based behaviors, including in the context of cocaine habits.

One proposed model for the manner in which the prelimbic cortex coordinates goal-directed action is via interactions with the dorsomedial striatum and basolateral amygdala[10]. Thalamic relays and direct basolateral amygdala inputs innervate deep-layer prelimbic cortical neurons[36, 37], so we reasoned that dendritic spine plasticity could be involved in action–outcome learning and memory. Supporting this perspective, individual differences in dendritic spine densities on layer V prelimbic cortical neurons correlated with response selection strategies: lower spine densities were associated with goal-directed responding, and higher densities with habit-based behavior.

Could remodeling prelimbic cortical dendritic spines enhance memory for action–outcome relationships? To test this possibility, we used the ROCK inhibitor fasudil. ROCK is a major substrate of the RhoA GTPase that phosphorylates LIM-kinase, which then phosphorylates cofilin. Cofilin typically cleaves F-actin, but phosphorylation inactivates it[17]. Thus, ROCK inhibition can facilitate the activity-dependent plasticity of dendritic spines, including dendritic spine pruning[26]. Fasudil is brain-penetrant, enhances learning and memory in other contexts[18–21], and is clinically approved for therapeutic use in Japan, with a positive safety profile[38].

Naïve mice were first trained to respond for food reinforcers such that they would be expected to develop habits by virtue of extensive response training. We then administered fasudil immediately following the modification of a familiar action–outcome contingency, during the presumptive consolidation of new knowledge (i.e., that a familiar behavior is no longer likely to be reinforced). Subsequently, fasudil-treated mice inhibited responding, whereas control mice persisted in responding, a habit-based strategy. In other words, fasudil enhanced action–outcome learning and memory, blocking habit-based behavior. Prelimbic cortical fasudil infusions had the same effects, whereas infusions into the anterior cingulate cortex had no significant consequence. This may be because the anterior cingulate cortex is preferentially involved in effort-based decision making and certain types of error detection[39]. Meanwhile, effort

requirements were not manipulated here, nor was an alternative response available during the contingency degradation procedure, potentially reducing anterior cingulate cortex involvement in task performance. Delaying fasudil treatment 4 or 18 h following training also had no effect, suggesting that ROCK inhibition enhances the consolidation, in particular, of action–outcome memory.

As predicted, fasudil also reduced dendritic spine densities on prelimbic cortical neurons. The consolidation of new memory is often assumed to involve the formation of new dendritic spines, but growing evidence indicates that dendritic spine pruning is also important[34, 35], potentially serving to enhance signal:noise during key learning opportunities. Although fasudil decreased dendritic spine density by only 9%, this value exceeds baseline dendritic spine turnover rates in the nearby motor and barrel cortices[40, 41]. Further, spine loss could be accounted for by a loss of mushroom-shaped spines, which are likely to be synapse containing. Even in the nearby anterior cingulate cortex, we detected a trend for reduced dendritic spine volume, consistent with the reduction of mushroom-shaped spines in the prelimbic cortex (Supplementary Table 1). By contrast, when fasudil was administered 1 day after conditioning, i.e., after potential memory consolidation, no changes were detected. Together, these findings suggest that ROCK inhibition prunes prelimbic cortical dendritic spines in an activity-dependent manner, facilitating subsequent goal-directed response choice.

We next determined whether fasudil could block habitual responding for cocaine. We endeavored to decrease the value of cocaine, with the hypothesis that fasudil would enhance behavioral sensitivity to devaluation. Devaluing cocaine has proven particularly challenging in the field, so we utilized an oral delivery approach previously applied to rats[1] and mice[42]. In this case, mice responded for a cocaine–sucrose solution, which was later paired with LiCl, inducing transient malaise and thus decreasing its value. Fasudil-treated mice subsequently reduced responding when returned to the conditioning chambers, whereas control mice failed to modify their behavioral patterns, responding habitually. Thus, fasudil reduced habit-based responding for cocaine. Orally delivered cocaine readily penetrates the brain[43], but one concern with this experimental design is that responding in fasudil-treated mice could have decreased because the value of the ingested sucrose, but not necessarily the cocaine, diminished. To address this possibility, we implanted indwelling jugular catheters, allowing mice to self-administer intravenous-delivered cocaine. Mice with a history of fasudil treatment responded less, confirming that fasudil mitigated cocaine, and not simply sucrose, habits.

Our experimental design can be envisioned as paralleling a common scenario in cocaine use disorders: (1) Mice self-administer cocaine in operant-conditioning chambers, whereas humans use cocaine in their daily lives. (2) As a consequence of the devaluation procedure, mice come to associate cocaine with an adverse outcome (malaise), whereas humans may seek treatment due to the negative consequences of cocaine abuse. (3) Despite adverse consequences, vehicle-treated mice continue to respond habitually for cocaine, whereas fasudil-treated mice reduce responding. After treatment in humans, 40–60% of patients relapse[44]. Pairing ROCK inhibitors with cognitive behavioral therapy in humans could be an effective pharmacological adjunct to reduce the rate of relapse. Indeed, pharmacological regulators of actin cytoskeleton dynamics may have broad potential in treating drug use disorders. It is widely acknowledged that cocaine modifies dendrite and dendritic spine structure in multiple brain regions (reviewed in ref. [45]). Cocaine also appears to alter actin function[46–48]. In addition, integrins (extracellular matrix protein receptors) and their ligands and intracellular substrates such as p190RhoGAP and Abl2/Arg impact cocaine seeking in rats and cocaine-induced locomotor sensitization and cognitive impairments in mice[30, 49–53]. Further, infusion of latrunculin A, which inhibits F-actin polymerization, or myosin IIB inhibition, into the basolateral amygdala can block methamphetamine-related memories[54, 55].

We show that the ROCK inhibitor fasudil reduces prelimbic cortical dendritic spine densities when paired with action–outcome conditioning, and that dendritic spine density correlates with subsequent response selection strategies, which we interpret as a remnant of activity-dependent spine pruning. To confirm that cellular structural plasticity is necessary for fasudil to enhance goal-directed action selection, we co-administered fasudil with latrunculin A, blocking F-actin polymerization. As expected, control mice responded for cocaine even following its devaluation—a habit—whereas fasudil reduced habitual responding for cocaine. Meanwhile, simultaneous latrunculin A infusion blocked this effect, supporting the perspective that fasudil enhances memory formation—blocking habits—via F-actin turnover in the prelimbic cortex. Interestingly, latrunculin A alone also reduced habitual responding for cocaine. We initially attributed this outcome to unintended leakage into the ventrally situated infralimbic cortex, necessary for the expression of habits[56]. However, when we empirically tested this possibility, selectively infusing latrunculin A into the infralimbic cortex, we discovered no obvious effects (not shown). An alternative possibility is that new learning requires a period of spine turnover or change, followed later by stabilization, and latrunculin A impacted this second stabilization phase. A conceptually similar model was recently described for prelimbic cortical ERK42/44 phosphorylation in exerting rapid and then delayed influences in action–outcome memory formation[57].

During his tenure, former NIMH director Dr. Tom Insel called for the development of statin-like compounds (such as ROCK inhibitors) to treat psychiatric disorders (e.g., ref. [58]). Given its favorable safety profile and our evidence that it can mitigate cocaine self-administration, fasudil is a strong candidate, with the caveats that we envision it administered as an adjunct to behavioral therapy and potentially during early phases of drug withdrawal. An additional challenge in the field is understanding at a deeper, mechanistic level how goal-directed and habitual behaviors are balanced. During the acquisition of a new task, deep-layer prelimbic cortical neurons generate high firing rates that then decrease with repetition and habit formation[59]. This gradual "quieting" could explain why extensive training produces insensitivity to changes in action–outcome relationships and outcome value. By extension, the re-engagement of goal-directed response strategies after habits have formed may require re-activation of these neurons. Our findings suggest that optimizing signal:noise via selective dendritic spine pruning could facilitate this process.

## Methods

**Subjects**. Mice were C57BL/6 wildtype from Jackson Labs, transgenic mice expressing *thy1*-YFP-H[60] and fully back-crossed onto a C57BL/6 background, or the offspring of *arg*+/− × *arg*+/− crosses[61]. *arg* mutant mice were maintained on a mixed 129 Sv/J × C57BL/6 background with genotypes confirmed by PCR. Behavioral training began on postnatal day ~42 throughout. Mice were male unless otherwise noted. All procedures were approved by the Emory or Yale institutional animal care and use committees, as appropriate.

**Intracranial surgery**. Mice were anesthetized with a 100 mg/kg ketamine/1 mg/kg xylazine mixture and placed in a stereotaxic frame. With infusion needles centered at bregma, a hole was drilled in the skull corresponding to +2.0 AP, +/−0.1 ML, −2.8 DV.

For viral vector experiments, AAV5-CaMKII-HA-hM4D(Gi)-IRES-mCitrine or AAV5-CaMKII-GFP (UNC Viral Vector Core) was infused bilaterally (0.5 μl per hemisphere) over 5 min, with the needle left in place for an additional 5 min. Mice

were sutured and allowed to recover for 3 weeks, allowing time for viral vector expression.

For pharmacological studies, 360 μM fasudil[62], 10 mM STI-571[30], or sterile saline (0.1 μl per hemisphere) was infused over a period of 1 min immediately following the behavioral test session. Latrunculin A or sterile saline (0.15 μl per hemisphere) was delivered in a concentration of 5 μg/μl over 2 min (adapted from refs. [30, 63]) 30 min following systemic injection of saline or fasudil. Needles were left in place for 2 additional min following infusion, and mice were sutured and allowed to recover for 3 days.

**Histology**. Mice were killed by rapid decapitation and brains were submerged in 4% paraformaldehyde for 48 h and then transferred to 30% w/v sucrose. 50 μm-thick sections were prepared on a microtome held at −15 °C ± 1. Sections were mounted and infusion terminals were documented following examination under magnification. In the case of DREADDs experiments, mCitrine or GFP was imaged. Mice with terminals or fluorescence outside of the prelimbic prefrontal cortex were either excluded or segregated into a separate comparator group (Fig. 3).

**Instrumental response training**. Mice were food restricted to 87–90% of their original body weight. Med-Associates operant-conditioning chambers equipped with nose-poke recesses and a food magazine were used. Mice were trained to respond on two recesses for food reinforcement (20 mg grain-based pellets; Bio-serv), with 30 pellets associated with each response (60 pellets per session). Mice were trained for 5–6 sessions using a fixed ratio one schedule (for the STI-571 infusion experiment), then additionally, two sessions at RI30 (all others). For extended training—i.e., to induce habit behavior in intact control mice—9 RI60 sessions followed. Sessions terminated when all pellets had been delivered or at 135 min.

Cocaine-reinforced experiments utilized a 75 μg/ml cocaine (Sigma) + 10% w/v sucrose solution delivered into the magazine by a motorized dipper that held 100 μl. A single response (nose-poke) was reinforced, and up to 30 reinforcers were delivered per session. The average daily dose of cocaine following the response acquisition phase was 12.5 mg/kg.

**Action–outcome contingency degradation**. Methods were consistent with prior investigations[20, 21, 42]. Briefly, on one day, one of the two nose-poke recesses was occluded, and pellet delivery was contingent upon responses on the remaining available nose-poke recess. A variable ratio 2 (VR2) schedule of reinforcement was used for 25 min. During another session, only the opposite nose-poke recess was available. In this case, pellets were delivered independently of animals' interactions with this nose-poke recess for 25 min. The pellet delivery rate was matched to each animal's reinforcement rate from the previous session (also as in ref. [64]). Thus, responding on one nose-poke recess became significantly less predictive of pellet delivery than the other (see ref. [5]). The location of these nose-poke recesses was counter-balanced.

The following day, both nose-poke recesses were available for 5–15 min during a probe test conducted in extinction. Preferential engagement of the previously highly-reinforced response is considered "goal-directed," whereas non-selective responding reflects habit-based behavior[27].

**Outcome devaluation**. After instrumental response training, mice were placed in a clean empty cage and allowed 1 h of free access to the reinforcer pellets or oral cocaine–sucrose solution used in the instrumental conditioning experiments. Immediately following, mice were injected with lithium chloride (LiCl) (0.15 M, 4 ml/100 g body weight, i.p.[65]), inducing temporary malaise and CTA. This pairing was repeated two or three times, as indicated in the figures (Figs. 2f, 5c, and 6d). The amount of food or liquid consumed was measured after each pairing.

For mice trained to respond for a cocaine–sucrose solution, a "reminder session" consisted of mice being then placed in the operant-conditioning chambers for 10 min, providing mice with an opportunity to update the association between the operant response and the now-devalued reinforcer[32, 33]. Saline or fasudil injection immediately followed this session, as indicated in the figure timelines.

For all mice, a probe test conducted in extinction then measured whether mice responded for the now-devalued reinforcer. Response rates were compared to the last day of training. Following the probe test, mice were given a "post-probe consumption test," consisting of 1 h of free access to the reinforcer, to re-confirm the acquisition of CTA. One mouse each in the initial oral cocaine (Fig. 5) and latrunculin A experiments (Fig. 6) did not reduce consumption, as measured between the first pairing and the post-probe consumption test, and were excluded.

**Intravenous catheter implantation**. Mice were anaesthetized with a 100 mg/kg ketamine/1 mg/kg xylazine mixture, and the dorsal and ventral sides were shaved and disinfected. The right jugular vein was exposed by blunt dissection, and a sterile Silastic catheter was placed[66] and then exteriorized posterior to the scapulae. The entrance and exit wounds were sutured, and mice were housed individually. During the 5–7 day recovery period, catheter patency was ensured by infusing mice daily with 0.05 ml heparinized saline. Subsequently, catheter patency was tested weekly using a 0.03 ml ketamine challenge (15 mg/ml). If mice were insensitive to

ketamine at any point, defined by a failure to lose muscle tone within 10 s of infusion, they were excluded.

**Intravenous cocaine self-administration**. Following catheter implantation, cocaine self-administration was tested in contextually distinct conditioning chambers relative to the chambers in which oral cocaine and food-reinforced test sessions had been conducted. Mice were tested daily, during which a single nose-poke response on the center of three nose-poke recesses was reinforced with an infusion of cocaine (20 μl; 1.25 mg/ml) delivered through a catheter connected to a swivel holding armored polyethylene tubing. Delivery culminated in extinction of the house light and a 20-s timeout. Sessions ended when mice self-administered 30 infusions or at 120 min. Mice were considered to have acquired the cocaine-reinforced response when they self-administered > 20 infusions (~20 mg/kg)[67].

**Extinction training**. Mice were placed in the conditioning chambers in which they had been trained to respond for food for 3 × 45-min sessions (1 per day), with no reinforcement delivered.

**Clozapine N-oxide delivery**. For DREADDs experiments, all mice, regardless of viral vector, received 1 mg/kg Clozapine N-oxide (CNO) (2% DMSO in sterile saline, i.p., 1 ml/100 g; Sigma) 1 h prior to the "degradation" training session. Mice were tested in the probe test in the absence of further injection. Response rates during this drug-free probe test are shown.

**Systemic fasudil delivery**. Overall, 10 mg/kg fasudil[20] (LC Labs), dissolved in sterile saline, was administered i.p. (1 ml/100 g) at the time points indicated in the experimental outlines.

**Dendritic spine imaging and analysis**. thy1-YFP-expressing mice (expressing YFP in layer V neurons) were killed by rapid decapitation at the time points indicated in the experimental outlines. Brains were submerged in 4% paraformaldehyde for 48 h and then transferred to 30% w/v sucrose. Overall, 50 μm-thick sections were prepared on a microtome held at −15 °C ± 1. Images were acquired on a Leica DM5500B microscope equipped with a spinning disk confocal (VisiTech International) and a Hamamatsu Orca R2 camera using a 100 × 1.4 NA objective. Z-stacks of dendritic segments were acquired using a 0.1 μm step size. 8 dendrites were acquired bilaterally from 8 independent neurons per animal. The location of the imaged segments within target regions was confirmed by zooming out to a low magnification. Care was taken to select second-order or higher apical dendrites of a known distance from the soma. Dendritic spines were sampled from dendrites 50–150 μm from soma in our initial experiments (Fig. 2) and then from distal segments within this same window for subsequent experiments (Fig. 4), given that distal dendrites are considered more labile and thus, likely to change following drug treatment. A single blinded experimenter generated all images.

**Dendritic spine reconstruction**. The FilamentTracer module of Imaris (Bitplane AG) was used according to ref. [68]: a dendritic segment 15–25 μm in length was drawn with the autodepth function. Dendritic spine head location was manually indicated, and FilamentTracer processing algorithms were used to calculate morphological parameters. Dendritic spines were classified as stubby, mushroom or thin according to ref. [69]. A single blinded individual quantified all dendritic spines within a single experiment, with inter-rater reliability ensured between two experimenters.

**Statistics**. Sample sizes were determined based on prior experiments using similar approaches[5, 21, 42, 67]. Except when groups were assigned by matching response rates, mice were randomized to groups. Response rates were compared by ANOVA with group, or group and session, as factors, with repeated measures as appropriate. Following significant interactions, Tukey's post-hoc comparisons were used. Response preference ratios were calculated by dividing non-degraded/degraded response rates and comparing group means by Mann–Whitney U tests. Food/cocaine ingestion was compared by ANOVA or t-test as appropriate. For dendritic spine densities, each mouse contributed a single value for comparison by ANOVA or t-test as appropriate. Correlations were analyzed by linear regression. Two mice in the local fasudil infusion experiment (Fig. 3) and two mice in the local latrunculin A experiment (Fig. 6) generated values > 2 standard deviations above the mean and were excluded. In the case of non-normal distributions, square root or arcsin transformations were used as appropriate (for ANOVAs), or Mann–Whitney U tests (in place of t-tests) were applied. Tests were two-tailed throughout, and $p < 0.05$ was considered significant.

Individual dendritic spine parameters are reported in Supplementary Table 1 and were compared by Kolmogorov–Smirnov tests. Owing to the considerable degree of power generated in these analyses, only $p < 0.001$ was considered significant.

Statistical analyses were performed using SigmaStat or SPSS.

**Data availability**. Data sets can be provided upon reasonable request.

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

## Acknowledgements

We thank A. Allen and Ms Hayley Arrowood for their considerable assistance. We also thank Dr R. Jude Samulski (UNC), Ms Courtni Andrews, and Ms Yong Yang for assistance with DREADDs experiments; Dr Anthony Koleske for *Abl2/arg* mutant mice; and Gourley lab members for manuscript critiques. This work was supported by the Emory Egleston Children's Research Center, NIH MH101477, DA015040, DA034808, DA036316, and DA036737. The Yerkes National Primate Research Center is supported by the Office of Research Infrastructure Programs/OD P51OD011132-53. The Emory Integrated Cellular Imaging Core is supported by an NINDS Core Facilities grant, P30NS055077.

## Author contributions

Experiments were designed and conducted by A.M.S. and S.L.G., except L.M.D. conducted the intravenous cocaine self-administration parts of experiments. A.M.S. and S.L.G. wrote the manuscript with review from L.M.D. A.M.S. performed statistical analyses.

## Additional information

**Competing interests:** The authors declare no competing financial interests.

