## [Peer Review File · Nature Communications]

Reviewers' comments:

Reviewer #1 (Remarks to the Author):

The authors clearly show that systemic and localized prelimbic injection of the drug fasudil (which is thought to act by inhibiting the cytoskeletal regulatory factor Rho-kinase) can impact performance in a devaluation paradigm when contingencies are degraded or paired with an aversive LiCl CTA treatment. The authors set up experiments in which behavior is likely to be habitual, defined by its insensitivity to devaluation, and infuse fasudil immediately after devaluation conditioning. This leads to 'rescue' of flexible responding in which there is a difference in degraded and non degraded responses (here also quantified as a non zero preference ratio for both sexes). They show that both systemic and local infusion of fasudil immediately after conditioning affects responding for food (Fig 2) and but does not work after 4 and 18 hour delays (Fig S1). They also show fasudil after CTA paired with cocaine reduces specific responding for the devalued oral sucrose-cocaine and later self administration of cocaine (Fig 4). They then follow up with an experiment in which local PrL fasudil can enhance cocaine-sucrose CTA and is blocked by co injection of latrunculin A. They also show arg kinase KO mice produce habitual responding in a paradigm that produces goal directed behavior in WT mice (S3). Structural changes in spine density suggest loss of spines in the PrL is associated with less habitual responding within the fasudil group. This is feasibly associated with remodeling and learning.

The possibility of enhancing flexible responding when behavior has become habitual (by any means) is exciting. The fact that fasudil has a favorable safety profile and may act in an activity dependent manner is also exciting. This drives my favorable response to this manuscript.

The local infusions, time course, KO experiments and latrunculin controls reflect a rigorous and multi-pronged test of the theory and mechanisms behind the drug effects. There is important data hidden in the supplemental figures. Fig S1 and S2 should be included in the main manuscript.

My major concern is that the writing and the figures are not clear enough.

The description of the devaluation procedure is confusing.

The transitions between experiments in the results are confusing.

The training paradigm and even the chamber should be illustrated for each experiment in detail in each figure.

The spine loss story is less important in my opinion than the behavioral story, but an important attempt to connect to the cytoskeleton function of ROCK. The results here are weaker. I remain curious why doesn't fasudil affect cingulate neuron spine density. Effects in behavior and spines are both null. Could there be a dose or targeting issue? Leaks from superficial cingulate injection? Or are cingulate pyramidal neurons different in way that should make them insensitive to fasudil? If the answer is actually activity related, could the activity dependent hypothesis be further tested? This is not necessary for this manuscript, however. Minor issue, Fig 3 spine density decrease shown seems exaggerated compared to the more modest quantification.

Reviewer #2 (Remarks to the Author):

The authors present important and interesting findings concerning how structural plasticity of layer 5 pyramidal neurons is important in the progression from goal-directed decision making to habitual responding for both a natural reward and cocaine by inhibiting the action of ROCK, an important mediator of actin cytoskeletal remodeling.

This paper is well thought out with many important controls, but there are a few concerns that should be addressed prior to publication.

1) response rates in figure 1 appear to be lower than other response rates in figure 2 and 3 and that 5-7 day period. What accounts for this variability?

2) In the Figure 1 DREADD inactivation occurs before the degradation but subsequent manipulations with ROCK inhibitors occur after, why this change in time of manipulation? What are the effects of DREADD inhibition of PL during the probe test?

3) Figure 2 j-m, the n per group is highly variable (4-12) and the n/a particular group is not given. Thus it is unclear if the lack of effect in ACC control region is due to lack of statistical power because of too few mice.

3) For spine densities, there is almost a 2 fold difference in basal levels in fig 2 and 3, what accounts for this dramatic difference?

Also, it would be useful for the authors to show the data for other spine parameters as cumulative probability plots which will allow for an assessment in any shift in the population, which may be indicative of selective changes in a particular morphological type of spine (thin, mushroom, stubby)

4) The authors use a *arg-/-* to show bidirectional effect of ROCK activity on the ability to shift from habitual to goal-direction decisions, however, the basal levels of responding in the KO are significantly diminished making it difficult to interpret these findings. Another method of increasing ROCK activity, either pharmacologically or with viral mediated overexpression, would help provide greater support for this claim.

5) The authors should provide spine morphology data for ACC, in cumulative probability form

6) what accounts for the increase in response on inactive port in fasudil group in figure S3B?

7) In fig. 5) the combination of fasudil & latrunculin A leads to a decrease in the overall responding making it hard to interpret if indeed latrunculin blocked the effects of fasudil. This is more confusing considering latrunculin and fasudil have similar behavioral effects when delivered separately. The authors explanation of this might be a leak of latrunculin into the IL is unsatisfactory and should be confirmed experimentally.

Reviewer #3 (Remarks to the Author):

Using multidisciplinary approaches, the authors report that inhibition of Rho-kinase (ROCK) with fasudil enhanced action-outcome memory in mice that were trained to respond in a habit-like manner after degradation of a sucrose reward. When paired with new learning, fasudil decreased dendritic spine density in the prelimbic cortex during consolidation of a memory. The effects of fasudil on different types of cocaine reinstatement were prolonged and depended on actin turnover. The discovery that an FDA-approved drug has the ability to switch habit-based activity to goal-directed behavior has enormous potential for treatment of addiction-related disorders. The study is innovative, well-designed, and well-executed; the manuscript is well-written, illustrated, and informative. Only minor points need attention.

1. The authors need to show a real image of CAMKII-driven hM4Di and its spread in medial prefrontal

cortex not just a schematic from Allen Brain Atlas.

2. What is the evidence that only CAMKII-expressing neurons in PFC expressed hM4Di in their hands?
3. Perhaps the modest 9% decrease in spine density in prelimbic cortex would be enhanced if the authors classified the spine subtypes into stubby, filopodia, mushroom based on approaches in the literature.
4. In the Discussion (p 17), please point out the limitation that fasudil would need to be administered to people with cocaine use disorder in a very restricted period of time after the last cocaine exposure to be effective.
5. In the Discussion (p 17), please moderate the sentence "We show that...fasudil eliminates prelimbic cortical dendritic spines....." Spines were decreased by 9% so decreased may be more accurate.
6. The fact that latrunculin alone decreased cocaine-seeking behavior needs a stronger explanation (p 18). Wouldn't diffusion away from the prelimbic cortex mask its block of fasudil effects?

Dear colleagues,

I thank you for the time you spent with our manuscript, for your insightful critiques, and for your enthusiasm. We have made several key modifications per your feedback, and we conducted additional experiments and analyses. *These are:*

1. Replicating the pharmacological experiments in the original fig.2j-m to increase group sizes (now in fig.3);
2. Locally infusing STI-571, an Abl-family kinase inhibitor, into the prelimbic prefrontal cortex as a pharmacological complement to our studies using *Ab2*^{-/-} mutant mice (fig.3);
3. Conducting additional analyses of 3D-reconstructed dendritic spine datasets to ascertain whether specific dendritic spine subtypes (mushroom, thin, stubby) were modified by various manipulations. These data complement the parametric and Kolmogorov-Smirnov population comparisons that were included in the initial manuscript draft.
4. We also made textual modifications aimed at increasing clarity, particularly in the transitions between experiments, in the Results section. We have additionally improved our discussions regarding the anterior cingulate cortex and the timing of our injections.
5. We have moved experiments originally reported in the Supplementary Materials into the main text, per the suggestion of R.1.
6. In the original Discussion section, we speculated that some unanticipated effects of latrunculin A (in the absence of fasudil) could be attributable to off-target leak into the infralimbic cortex. R.2 encouraged us to test this hypothesis. We undertook this ambitious and technically demanding experiment, utilizing a full 2x2 experimental design to compare the effects of latrunculin A infusion into the prelimbic vs. infralimbic cortices on habit-based responding for cocaine.

Despite our efforts (we conducted the experiment twice), we generated no support for our hypothesis. We describe these efforts in the revised Discussion. We also pose an alternative model based on a recent report by Balleine *et al.*, who discovered two “waves” of ERK42/44 phosphorylation in the prelimbic cortex associated with action-outcome learning and memory.

Notably, points #1, 2, and 6 represent new experiments involving long-term training to induce habit-based behavior, *in vivo* intracranial surgery, histological confirmation of infusion sites, and then, replication. We believe that the results of our efforts as a whole strengthen our conclusions, and we thank the reviewers for their insightful suggestions and feedback.

As the reviewers might remember, an experiment in fig.1 utilizes Designer Receptors Exclusively Activated by Designer Drugs (DREADDs). The reviewers are no doubt aware of the recent report in *Science* (Gomez et al., 2017) suggesting that DREADDs are activated not by clozapine-N-oxide (CNO), but by its metabolite, clozapine. While this report provides an important reminder to use necessary control conditions, our experimental design included key controls from the outset. These are: 1) All mice in our experiment, regardless of viral vector group, received CNO. Thus, any unintended effects of CNO were experienced by all animals. 2) CNO was delivered in conjunction with action-outcome contingency degradation training, and response preferences were tested the following day, when mice were drug-free. These points are clear in the Methods section.

For these reasons, we have elected to retain our DREADDs experiment in this revision, despite the report of Gomez et al. If the reviewers have concerns regarding our decision, however, I hope they will give us the opportunity to consider them, given that this experiment is ultimately a small part of the overall story. For example, its omission would not affect the primary conclusion that Rho-kinase inhibition can strengthen action-outcome memory consolidation, improving goal-directed response choice.

Below, you will find a point-by-point response to critiques. Your comments are in bold text, while our responses are in italicized text. Per the Editor's request, areas in the primary document that were modified are highlighted in yellow. (Note also that figure 2 has been split into 2 separate figures due to inclusion of new data.)

Reviewers' comments

Reviewer #1 (Remarks to the Author):

The authors clearly show that systemic and localized prelimbic injection of the drug fasudil (which is thought to act by inhibiting the cytoskeletal regulatory factor Rho-kinase) can impact performance in a devaluation paradigm when contingencies are degraded or paired with an aversive LiCl CTA treatment. The authors set up experiments in which behavior is likely to be habitual, defined by its insensitivity to devaluation, and infuse fasudil immediately after devaluation conditioning. This leads to 'rescue' of flexible responding in which there is a difference in degraded and non degraded responses (here also quantified as a non zero preference ratio for both sexes). They show that both systemic and local infusion of fasudil immediately after conditioning affects responding for food (Fig 2) and but does not work after 4 and 18 hour delays (Fig S1). They also show fasudil after CTA paired with cocaine reduces specific responding for the devalued oral sucrose-cocaine and later self administration of cocaine (Fig 4). They then follow up with an experiment in which local PrL fasudil can enhance cocaine-sucrose CTA and is blocked by co injection of latrunculin A. They also show arg kinase KO mice produce habitual responding in a paradigm that produces goal directed behavior in WT mice (S3). Structural changes in spine density suggest loss of spines in the PrL is associated with less habitual responding within the fasudil group. This is feasibly associated with remodeling and learning.

The possibility of enhancing flexible responding when behavior has become habitual (by any means) is exciting. The fact that fasudil has a favorable safety profile and may act in an activity dependent manner is also exciting. This drives my favorable response to this manuscript.

The local infusions, time course, KO experiments and latrunculin controls reflect a rigorous and multi-pronged test of the theory and mechanisms behind the drug effects. There is important data hidden in the supplemental figures. Fig S1 and S2 should be included in the main manuscript.

Thank you for your support for this work. I have shifted the delayed injection and arg-/- experiments to the main text as you suggest (new fig.2 & 3, respectively). R.2 raised a concern regarding the arg-/- mutant mice, so we conducted an additional experiment in which we selectively inhibited Arg in the prelimbic cortex pharmacologically. Consistent with experiments using arg-/- mice and with the document as a whole, site-selective Arg inhibition caused habit-based behaviors. Per your suggestion, we include this experiment in the main document alongside the arg-/- experiment.

My major concern is that the writing and the figures are not clear enough.

- **The description of the devaluation procedure is confusing.**
- **The transitions between experiments in the results are confusing.**
- **The training paradigm and even the chamber should be illustrated for each experiment in detail in each figure.**

Thank you for these notes. We have thoroughly revised our Results section, with a particularly careful eye towards clarifying the transitions between the experiments. Throughout, additional explanations of the experimental procedure should aid the reader in understanding what each

assay tests, what the hypotheses were, and what the outcomes were. We also took special care to clarify the devaluation procedure at its first description.

Secondly, we have added illustrations of the operant conditioning chambers to figures 1-5, per the reviewer's suggestion. (We did not add an illustration to fig.6 because it would be the same as fig.5, noted in the caption.) These illustrations complement the experimental timelines, and they also illustrate the training approach because they illustrate which nose poke ports are active, what the reinforcer is and how it is delivered, etc. We thank the reviewer for this opportunity to improve the accessibility of our document.

The spine loss story is less important in my opinion than the behavioral story, but an important attempt to connect to the cytoskeleton function of ROCK. The results here are weaker. I remain curious why doesn't fasudil affect cingulate neuron spine density. Effects in behavior and spines are both null. Could there be a dose or targeting issue? Leaks from superficial cingulate injection? Or are cingulate pyramidal neurons different in way that should make them insensitive to fasudil? If the answer is actually activity related, could the activity dependent hypothesis be further tested? This is not necessary for this manuscript, however.

Thank you for these points. With regards to dosing, it is possible that a higher concentration of fasudil infused into the cingulate would modify behavioral outcomes, but we favor the interpretation that the cingulate is not required for optimal responding in this particular task. We more fully explain our thinking in the revised Discussion.

In response to the reviewer's comment, we considered, but ultimately rejected, other possibilities: These included regionally different expression patterns of ROCK, after finding no evidence that protein levels differ between the cingulate and prelimbic cortices. Another consideration was that the cingulate infusions in our original manuscript were caudal to the prelimbic infusions; thus, we thought that it may simply be that ROCK in the anterior rather than posterior medial prefrontal cortex is important for action-outcome decision making (rather than prelimbic vs. cingulate). However, when we replicated our fasudil infusion experiment for this revision, we inadvertently generated cingulate-selective infusions in the rostral cingulate, with no effect. This brings us back to our original hypothesis regarding activity dependence, which we will assess in future experiments.

With regards to effects of fasudil on dendritic spines: Based on the comments of the other reviewers, we further probed our dendritic spine datasets, revealing two additional findings of note regarding ACC neurons: Firstly, dendritic spine subtype densities were not affected by fasudil (as with overall densities). Nevertheless, population analyses revealed a modest decrement in dendritic spine volume following fasudil in the ACC. This effect, now reported in Supplementary Table 1, is consistent with fasudil effects in the prelimbic cortex, where a reduction in mushroom-shaped dendritic spines was observed. Thus, fasudil does not fully spare ACC neurons; rather, it has less pronounced effects relative to prelimbic cortical neurons. We have noted this in our revised Discussion section, and we thank the reviewer for encouraging us to think more about this issue.

Minor issue, Fig 3 spine density decrease shown seems exaggerated compared to the more modest quantification.

Thank you for this point. In response to R.3, we classified dendritic spines, revealing a loss of mushroom-shaped spines in this group. We have added these data to the figure, and we feel

that the illustrative dendrite is more in line with this finding. We have textually clarified in the caption that this dendrite is from a 'high-responding' mouse. We thank the reviewer for the opportunity to clarify this point.

Reviewer #2 (Remarks to the Author):

The authors present important and interesting findings concerning how structural plasticity of layer 5 pyramidal neurons is important in the progression from goal-directed decision making to habitual responding for both a natural reward and cocaine by inhibiting the action of ROCK, an important mediator of actin cytoskeletal remodeling. This paper is well thought out with many important controls, but there are a few concerns that should be addressed prior to publication.

1) response rates in figure 1 appear to be lower than other response rates in figure 2 and 3 and that 5-7 day period. What accounts for this variability?

I believe the reviewer is referring to the response rates during sessions 6-7 when the schedule of reinforcement shifts from a fixed ratio to a random interval. Unfortunately, this is likely reflective of two different experimenters, one less strict with food restriction than the other (our goal range is 87-90% of baseline body weight). We agree with the reviewer's implication that the acquisition curve is distracting, given that the point of this figure is to replicate in mice the finding from rat studies that inactivation of the prelimbic cortex causes a deferral to habit-based behavior in a contingency degradation assay (Balleine and Dickinson, 1998). As such, we have shifted the acquisition curve to fig.S1.

2) In the Figure 1 DREADD inactivation occurs before the degradation but subsequent manipulations with ROCK inhibitors occur after, why this change in time of manipulation? What are the effects of DREADD inhibition of PL during the probe test?

With the DREADDs experiment, we were motivated by Ostlund and Balleine (2005), who reported that prelimbic cortex inactivation prior to reinforcer devaluation, but not at the probe test, interferes with action-outcome decision making in rats. Hence, the prelimbic cortex is thought to be involved in the formation, but not expression, of action-outcome learning and memory, and we based our inactivation timing (prior to contingency degradation) accordingly.

We then hypothesized that the prelimbic cortex is specifically involved in the consolidation of action-outcome conditioning. Consolidation is best manipulated using injections that immediately follow training because they would not affect the acquisition (before) or expression (later) phases of the experiment. We more carefully explain our rationale and the context for these experiments in our revision (Introduction and Results). We thank the reviewer for opportunity to more explicitly describe our thinking.

3) Figure 2 j-m, the n per group is highly variable (4-12) and the n/a particular group is not given. Thus it is unclear if the lack of effect in ACC control region is due to lack of statistical power because of too few mice.

Thank you for this comment. The reviewer is correct that the smallest group was the ACC group. In response to this comment, we generated a small replication experiment, explicitly targeting the ACC, which increased the group sizes. Again, we found no effect of infusion. The

graph and associated text have been updated accordingly, and we have also indicated exact n/group.

3) For spine densities, there is almost a 2 fold difference in basal levels in fig 2 and 3, what accounts for this dramatic difference?

Thank you for this point. In fig.2, brains were collected 3 days after injection, while in fig.3 (now fig.4), brains were collected 1 hour after injection. Thus, one contributing factor may be acute injection stress, which can cause rapid dendritic spine proliferation on apical dendrites in the prelimbic cortex (Seib and Wellman, 2003). We have noted this possibility in our revised Results section.

Additionally, the reviewer's comment highlighted for us a regrettable lack of clarity in our Methods, which we have corrected. Specifically, our imaging approach initially focused on dendrites 50-150 um from the soma (fig.2), but then in subsequent studies (fig.4), we preferentially imaged in the distal regions of the same window since distal dendrites are considered more labile and thus, likely to change following drug treatment. This has been clarified in the Results and Methods, and we appreciate the reviewer pointing out this issue.

Also, it would be useful for the authors to show the data for other spine parameters as cumulative probability plots which will allow for an assessment in any shift in the population, which may be indicative of selective changes in a particular morphological type of spine (thin, mushroom, stubby)

Thank you for this point; we agree. We are sorry that we were not clear in our initial submission. We used Kolmogorov-Smirnov comparisons in our original analyses to test for differences in distribution shape. As the reviewer knows, the K-S test is designed to assess differences in distribution, which would be shown in cumulative density plots. Because the K-S tests were non-significant, we elected not to show those graphs, but we do provide the outcomes of our K-S comparisons in Supplementary Table 1. Notably, we found no group differences even when each spine, rather than each dendrite, was considered an independent sample, which increases statistical power in this already highly-powered test.

Both you and R.3 urged us to further probe our dendritic spine datasets. In response, we classified dendritic spines as "mushroom," "stubby," or "thin" according to established parameters (Radley et al., 2013). This analysis revealed selective pruning of mushroom-shaped spines in the prelimbic cortex when fasudil was paired with new learning. These data have been added to fig.4, and they strengthen our conclusions. In supplementary fig.4, we have also added the results of spine classifications for the other conditions, revealing no differences between groups.

4) The authors use a arg-/- to show bidirectional effect of ROCK activity on the ability to shift from habitual to goal-direction decisions, however, the basal levels of responding in the KO are significantly diminished making it difficult to interpret these findings. Another method of increasing ROCK activity, either pharmacologically or with viral mediated overexpression, would help provide greater support for this claim.

Thank you for this point. We generated a new experiment using genetically intact mice administered local infusions of STI-571, an inhibitor of Abl-family kinases (Abl and Arg). We administered STI-571 immediately following instrumental contingency degradation (as with

fasudil infusions), inducing habit-based behavior, as would be expected. The data have been integrated into main text fig.3 and supplementary fig.3.

Notably, response rates were overall lower following STI-571 infusion, as with *arg-/-* mice. This is consistent with the effects of prelimbic cortical lesions in rats (Corbit and Balleine, 2003) and mice responding on a progressive ratio schedule (Gourley et al., 2010). We address these changes in overall response rates in the revised Supplementary Materials document.

5) The authors should provide spine morphology data for ACC, in cumulative probability form

Thank you for this suggestion. These data and comparisons were added to Supplementary Table 1. Notably, we detected one trend-level effect, graphed below:

This effect is in general agreement with our other findings (smaller spine volume is consistent with the gross reduction in mushroom-shaped spines in the nearby prelimbic cortex). This has been noted in the revised Discussion.

Like the other population comparisons in this report, this one is not overwhelming, so we have elected not to include graphical representations of these data in the manuscript. If the reviewer strongly opposes this decision, however, we would be willing to include them.

6) what accounts for the increase in response on inactive port in fasudil group in figure S3B?

This is unclear. Here, the mice were first trained to respond for an orally-ingested cocaine reinforcer by nose-poking. We then devalued the cocaine and shifted them to a novel context for sucrose pellet self-administration (also nose-poking), with the goal of confirming that responding for a non-devalued reinforcer would remain robust. Even though the chambers were contextually different, both had inactive nose poke apertures. Thus, mice presumably had some concept of an inactive response. Why the fasudil-treated mice would seem to be more willing to sample an inactive port is unclear. We have indicated in the caption of this figure (now fig.S5b) that context A and context B in this experiment are the same as shown in the primary text associated with fig.5, where cartoon representations of the two contexts are now provided, per the request of R. 1. Although this does not answer the reviewer's question, we hope that this clarification of our experimental design might provide full transparency for readers who may develop hypotheses regarding this unexpected (though statistically non-significant) effect.

7) In fig. 5) the combination of fasudil & latrunculin A leads to a decrease in the overall responding making it hard to interpret if indeed latrunculin blocked the effects of fasudil.

Thank you for this comment – we regret the ambiguity in our figure. The response rates following fasudil/lat.A are actually compared to the last day training, when mice were drug naïve. Thus, fas+latrA is not reducing response rates. Rather, the mice assigned to the fas+latrA group were responding at slightly lower rates before any drug. This occurred by chance (becoming clear after the exclusion of mice with mis-targeted infusions).

Nevertheless, the reviewer's comment alerted us to the possibility that other readers could come away with the same mis-impression that fas+latrA reduces overall response rates. We thus generated a small replication experiment with higher-responding mice, improving the matching of pre-drug response rates. The figure, caption, and statistics have been updated accordingly, and we thank the reviewer for pointing out this concern.

This is more confusing considering latrunculin and fasudil have similar behavioral effects when delivered separately. The authors explanation of this might be a leak of latrunculin into the IL is unsatisfactory and should be confirmed experimentally.

Yes, we agree – this was a surprising effect that was also highly replicable (we conducted this experiment 4 times). R.3 also encouraged us to question this finding further. As discussed in our cover letter, we attempted to generate support for our speculation, but with no positive outcomes, i.e., latrunculin A infusion into the infralimbic cortex had no affect. We have revised our Discussion to indicate as much. We also now discuss a new study from Balleine and colleagues describing 2 waves of ERK42/44 phosphorylation in the prelimbic cortex associated with new action-outcome learning. The second wave corresponds with the timing of latrunculin A infusions here. Given that ERK42/44 can be associated with dendritic spine stability, it may be that interactions between latrunculin A and ERK42/44 (in the absence of fasudil) can account for the unexpected effect of latrunculin A. Future experiments utilizing ERK42/44 inhibitors could test this hypothesis, but we felt they were beyond the scope of the current manuscript.

Reviewer #3 (Remarks to the Author):

Using multidisciplinary approaches, the authors report that inhibition of Rho-kinase (ROCK) with fasudil enhanced action-outcome memory in mice that were trained to respond in a habit-like manner after degradation of a sucrose reward. When paired with new learning, fasudil decreased dendritic spine density in the prelimbic cortex during consolidation of a memory. The effects of fasudil on different types of cocaine reinstatement were prolonged and depended on actin turnover. The discovery that an FDA-approved drug has the ability to switch habit-based activity to goal-directed behavior has enormous potential for treatment of addiction-related disorders. The study is innovative, well-designed, and well-executed; the manuscript is well-written, illustrated, and informative. Only minor points need attention.

1. The authors need to show a real image of CAMKII-driven hM4Di and its spread in medial prefrontal cortex not just a schematic from Allen Brain Atlas.

Thank you for this suggestion. We have added a representative coronal section to main text fig.1.

2. What is the evidence that only CAMKII-expressing neurons in PFC expressed hM4Di in their hands?

We have not tested this. Given the reviewer's question, we have carefully re-assessed our document to ensure that we are not using language to suggest as much. Thank you for this opportunity to clarify our language.

3. Perhaps the modest 9% decrease in spine density in prelimbic cortex would be enhanced if the authors classified the spine subtypes into stubby, filopodia, mushroom based on approaches in the literature.

Thank you for this suggestion. We classified spines, revealing a 16% deficit in mushroom-shaped spines, a significant loss. No such modifications were identified when fasudil injection was uncoupled from the opportunity for new learning. These new data have been added to fig.4 and supplementary fig.4. We used the classification parameters of Radley et al., 2013, and this reference has been added to the Methods accordingly.

4. In the Discussion (p 17), please point out the limitation that fasudil would need to be administered to people with cocaine use disorder in a very restricted period of time after the last cocaine exposure to be effective.

Yes, we agree that this is an important point, and we have emphasized it in our concluding paragraph.

5. In the Discussion (p 17), please moderate the sentence “We show that...fasudil eliminates prelimbic cortical dendritic spines.....” Spines were decreased by 9% so decreased may be more accurate.

Yes, we again agree and have modified this section accordingly.

6. The fact that latrunculin alone decreased cocaine-seeking behavior needs a stronger explanation (p 18). Wouldn't diffusion away from the prelimbic cortex mask its block of fasudil effects?

Yes, we agree – this was a surprising effect that was also highly replicable (we conducted this experiment 4 times). R.2 also encouraged us to question this finding further. As discussed in our cover letter, we attempted to generate support for our speculation that diffusion into the infralimbic cortex could impact cocaine seeking, but with no positive outcomes. I.e., latrunculin A infusion into the infralimbic cortex did nothing. We have revised our Discussion to indicate as much. We also now discuss a new study from Balleine and colleagues describing 2 waves of ERK42/44 phosphorylation in the prelimbic cortex associated with new action-outcome learning. The second wave corresponds with the timing of latrunculin A infusions here. Given that ERK42/44 can be associated with dendritic spine stability, it may be that interactions between latrunculin A and ERK42/44 (in the absence of fasudil) can account for the unexpected effect of latrunculin A. Future experiments utilizing ERK1/2 inhibitors could test this hypothesis, but we felt they were beyond the scope of the current manuscript.

REVIEWERS' COMMENTS:

Reviewer #1 (Remarks to the Author):

I am satisfied by the changes made and have no further comments.

Reviewer #2 (Remarks to the Author):

Swanson et al have nicely revised their important and interesting finding concerning how plasticity of distinct PFC neurons PrL vs ACC regulates the transition between goal-directed and habitual action selection. The text and figures are presented with greater clarity in regards to the rationale and experimental design. In addition, the reanalyze of the dendritic spine studies now highlights some important, previously missed findings. Primarily that the pruning of mushroom spines, likely to be the highly stable spines involved in habitual responding, occurred with new learning enabled by fasudil treatment. Also, interesting to see that there was a small shift in spine diameter in the fasudil group for ACC that is congruent with the findings of fasudil's effect on PrL spines. The control for dendrite diameter strengthens the findings.

Furthermore, their findings are significantly strengthened by the inclusion of the STI-571 infusions. While the response rates for this group are also lower than the controls with both non-degraded and degraded for STI-571 reaching the same level as degraded for the controls, this may be due to the challenging technical nature of the infusion experiments. Reassuringly there was no difference between the degraded and non-degraded ports resulting in likely habitual responding as indicated by the preference score.

Finally, their discussion of the surprising results of increased goal-directed behavior with latrunculin A provides satisfactory and plausible reasons for this finding. Subsequent studies probing this result will be very interesting although not necessary for this manuscript. As well, the discussion of the lack of effects of fasudil infusion into the ACC is much strengthened in this revision, along with the replication of this experiment which still showed no effect.

Overall, this manuscript is much improved and provides important insight into the molecular mechanisms underlying this shift in behavior that could be useful in the treatment of addictive-like disorders.

Reviewer #3 (Remarks to the Author):

The authors have satisfactorily responded to the previous critiques and have thereby strengthened the manuscript.

Point-by-point response to reviewers

None necessary.

(Reviewers' comments were:

Reviewer #1 (Remarks to the Author):

I am satisfied by the changes made and have no further comments.

Reviewer #2 (Remarks to the Author):

Swanson et al have nicely revised their important and interesting finding concerning how plasticity of distinct PFC neurons PrL vs ACC regulates the transition between goal-directed and habitual action selection. The text and figures are presented with greater clarity in regards to the rational and experimental design. In addition, the reanalyze of the dendritic spine studies now highlights some important, previously missed findings. Primarily that the pruning of mushroom spines, likely to be the highly stable spines involved in habitual responding, occurred with new learning enabled by fasudil treatment. Also, interesting to see that there was a small shift in spine diameter in the fasudil group for ACC that is congruent with the findings of fasudil's effect on PrL spines. The control for dendrite diameter was strengthens the findings.

Furthermore, their findings are significantly strengthen by the inclusion of the STI-571 infusions. While the responses rates for this group are also lower than the controls with both non-degraded and degraded for STI-571 reaching the same level as degraded for the controls, this may be due to the challenging technical nature of the infusion experiments. Reassuringly there was no difference between the degraded and non-degraded ports resulting in likely habitual responding as indicated by the preference score.

Finally, their discussion of the surprising results of increased goal-directed behavior with latrunculin A provides satisfactory and plausible reasons for this finding. Subsequent studies probing this result will be very interesting although not necessary for this manuscript. As well, the discussion of the lack of affects of fasudil infusion into the ACC is much strengthened in this revision, along with the replication of this experiment which still showed no effect.

Overall, this manuscript is much improved and provides important insight into the molecular mechanisms underlying this shift in behavior that could be useful in the treatment of addictive-like disorders.

Reviewer #3 (Remarks to the Author):

The authors have satisfactorily responded to the previous critiques and have thereby strengthened the manuscript.)